# Improvement of Survival over Time for Colorectal Cancer Patients: A Population-Based Study

**DOI:** 10.3390/jcm9124038

**Published:** 2020-12-14

**Authors:** Audrius Dulskas, Vytautas Gaizauskas, Inga Kildusiene, Narimantas Evaldas Samalavicius, Giedre Smailyte

**Affiliations:** 1Department of Abdominal and General Surgery and Oncology, National Cancer Institute, 1 Santariskiu Str., LT–08406 Vilnius, Lithuania; 2Institute of Clinical Medicine, Faculty of Medicine, Vilnius University, LT-03101 Vilnius, Lithuania; narimantas.samalavicius@gmail.com; 3Department of Abdominal Surgery, Vilnius University Hospital Santaros Clinics, 2 Santariskiu Str., LT-08661 Vilnius, Lithuania; gvytautas@gmail.com; 4Laboratory of Cancer Epidemiology, National Cancer Institute, LT-08406 Vilnius, Lithuania; inga.kildusiene@gmail.com (I.K.); giedre.smailyte@nvi.lt (G.S.); 5Department of Surgery, Klaipeda University Hospital, 41 Liepojos Str., LT-92288 Klaipeda, Lithuania; 6Health Research and Innovation Science Center, Faculty of Health Sciences Klaipeda University, 84 Herkaus Manto Str., LT-92294 Klaipeda, Lithuania; 7Institute of Health Sciences, Faculty of Medicine, Vilnius University, LT-03101 Vilnius, Lithuania

**Keywords:** colorectal cancer, colorectal cancer survival, national burden, Lithuanian cancer registry

## Abstract

Purpose: In this study, we analyzed the mortality and survival of colorectal cancer patients in Lithuania. Methods: This was a national cohort study. Population-based data from the Lithuanian Cancer Registry and period analyses were collected. Overall, 20,980 colorectal cancer patients were included. We examined the changes in colorectal cancer mortality and survival rates between 1998 and 2012 according to cancer anatomical sub-sites and stages. We calculated the 5-year relative survival estimates using period analysis. Results: Overall, 20,980 colorectal cancer cases reported from 1998 to 2012 were included in the study. The total number of newly diagnosed colorectal cancers increased from 1998–2002 to 2008–2012 by 12.1%. The highest number of colorectal cancers was localized and increased from 33.9% to 42.0%. The number of cancers with regional metastases and advanced cancers decreased by 11.1% and 15.5%, respectively. An increased number of new cases was observed for almost all colon cancer sub-sites. The overall 5-year relative survival rate increased from 37.9% in 1998–2002 to 51.5% in 2008–2012. We showed an increase in survival rates for all stages and all sub-sites. In the most recent period, patients with a localized disease had a 5-year survival rate of 78.6%, while survival estimates for advanced cancer patients remained low at 6.6%. Conclusion: Although survival rates variated in colorectal cancer patients according to disease stages and sub-sites, we showed increased survival rates for all patients.

## 1. Introduction

Worldwide, colon and rectal cancer is the third most commonly diagnosed cancer in males and the second in females [1]. Moreover, colorectal cancer is the second leading cause of cancer death in both sexes globally [1]. Its burden is expected to increase by 60% with more than 2.2 million new cases and 1.1 million cancer deaths by 2030 [2]. There is no effective primary prevention for colorectal cancer, but well-timed diagnosis, adequate staging and treatment can provide an excellent prognosis. A screening program allows doctors to diagnose the disease at earlier stages, thereby decreasing the mortality rate [3]. It is stated that colorectal cancer mortality decreased by 39.8% in men and 38.8% in women in the United States over the two last decades. In all 27 European Union member states, the decrease was slightly less: 13.0% and 27.0%, accordingly [2]. In addition, the highest colorectal cancer mortality rates were in Central and Eastern Europe (15.2 per 100,000), compared to the lowest rates in South-Central Asia (3.6 per 100,000) and Polynesia (4.4 per 100,000) [4]. Survival and mortality rates are different among countries, and survival rates have improved with time [5]. In order to assess the real life changes in cancer prevalence and to audit the diagnostic and treatment efficiency, cancer registries are the best tools [6].

A colorectal cancer screening program started in Lithuania in 2009 (in the first two largest cities with a population constituting 35% of the whole country). The program consists of performing an occult blood test (iFOBT) every two years starting at age 50 (until age 74) and a colonoscopy if the latter is positive. Since 2014, the program has been run nationwide. The program coverage is around 50% of the population.

However, there is limited detail and homogenously collected data about colorectal cancer in Lithuania, so our present study aimed to analyze the survival of colorectal cancer patients from 1998 to 2012 in the Lithuanian population. 

## 2. Materials and Methods

Ethical approval for the analysis of the population-based cancer registry data was not required.

The study was based on the Lithuanian Cancer Registry database, covering a population of less than 3 million residents, according to the 2018 census data. This database is a nationwide cancer registry containing personal, demographic and clinical information of all cancer patients in Lithuania since 1978. The registry is a part of the Cancer Incidence in Five Continents, which is a source of population-based cancer registries from all over the world [7]. In addition, it is possible to withdraw the death certificate for living status verification. The last date that a vital status assessment was conducted was on December 31, 2012. All data were anonymized before analysis.

In this study, we analyzed the survival of colorectal cancer (ICD-10 C18–C20, excluding large bowel lymphoma) patients diagnosed between 1998–2012. Cancer localization was assessed using the International Classification of Diseases ICD-O-3. We analyzed all the primary colorectal cancers. Aggregated sub-sites included the caecum and appendix (C18.0, C18.1); the right colon (C18.2, C18.3); the transverse colon (C18.4); the left colon (C18.5, C18.6, C18.7); other (C18.8, C18.9); and the rectosigmoid (C19) and rectum (C20).

Cancer stages drafted from the Lithuanian Cancer Registry were divided into three groups: (1) localized cancer T1-4/N0/M0; (2) advanced cancer with regional metastasis without distant spread: any T/N+/M0; and (3) advanced cancer with distant spread: any T/any N/M+ [8]. When patients underwent surgical treatment, pathological staging was given. For clinical staging, a computed tomography scan of the chest, abdomen and pelvis was used or a chest x-ray with an ultrasound. For rectal cancer, pelvic magnetic resonance imaging was performed. If any of these were missing, the data were termed as “missing data”.

Five-year relative survival estimates were calculated using period analysis. It provided more up-to-date survival estimates [9] Relative survival was calculated based on the ratio of the observed survival of cancer patients and the expected survival of the underlying general population. The latter was calculated according to the Ederer II method, using national life tables for the Lithuanian population stratified by age, gender and calendar year. The relative survival was adjusted for age using the international standard for cancer survival analysis. We performed the survival calculations with the freely available Stata statistical program [10].

## 3. Results

Overall, we included 20,980 colorectal cancer cases drafted from the Lithuanian Cancer Registry from 1998 to 2012. Table 1 shows cancer cases according to different time periods, cancer cases by sub-site and the extent of the disease. It also shows the differences between the time periods.

The total number of newly diagnosed colorectal cancers increased from 1998–2002 to 2008–2012 by 12.1% (from 5207 to 5950 cases, respectively). Most of colorectal cancers were localized and increased from 33.9 to 42.0%. The number of cancers with regional metastases and advanced cancers decreased by 11.1% and 15.5%, respectively. An increased number of new cases was observed for almost all colon cancer sub-sites, with little change in the number of rectosigmoid and rectal cancers.

The overall 5-year relative survival rate increased from 37.9% in 1998–2002 to 51.5% in 2008–2012 (Table 2). Better survival rates were seen for all disease stages and all sub-sites (Figure 1, Figure 2 and Figure 3). In the most recent period, patients with a localized disease had a 5-year survival rate of 78.6%, while survival estimates for advanced cancer remained low at 6.6%. Survival rates varied little by sub-site. The 5-year relative survival rate was slightly over 50% for colon cancer patients and almost 50% for rectal cancer patients in the period between 2008-2012.

## 4. Discussion

This is the first study to assess colorectal cancer survival in Lithuania over three different time periods by stage at diagnosis and cancer sub-site. Here, we showed that survival rates have improved significantly in Lithuania in colorectal cancer patients. There are many possible explanations for this improvement. First of all, the introduction of a regional bowel screening program (fecal occult blood test (iFOBT) performed every two years and a colonoscopy if the latter is positive) in Lithuania in 2009 (started in the capital city) may explain the slight increase in colorectal cancer incidence. The 5-year colorectal cancer survival rate in Lithuania has increased modestly over the past 14 years and is expected to increase further because of the screening program implemented nationwide in 2014. This national program has resulted in earlier diagnosis, possibly better treatment and survival options. Moreover, a further screening program test through a colonoscopy is important and its benefits are complex. An endoscopy is supposed to diagnose colorectal cancer earlier, but it also prevents cancer by removing the precursor lesions or adenomas [11].

It is important to note that, in the late 1990s, Lithuania changed the health system funding from local budgets (when it was very hard for a patient to reach a tertiary hospital center without paying additional money, and not all patients received proper health care) to a mixed system, mainly funded by the National Health Insurance Fund (NHIF). The Lithuanian health care system is thought to serve the whole population, and all residents have to pay for the health insurance scheme (typically paying 6–9% of taxable income). All residents pay the same amount of money and all should receive free health care [12].

Other important factors that may improve survival are minimal invasive surgical techniques, better perioperative care, the centralization of hospitals that operate on colorectal cancer and multidisciplinary team meetings for patients with advanced stages [13,14,15]. In addition to this, colorectal specialists, compared to when general surgeons performed surgeries, treat a higher numbers of colonic resections. Another important factor is the wider use of neoadjuvant chemo-radiotherapy in advanced rectal cancer [16,17,18,19,20]. Adjuvant chemotherapy in stages II and III of colon cancer has shown to increase survival by 5–15% in randomized trials [21]. Nevertheless, complete mesocolic excision and central vascular ligation have been emphasized recently, although data are opposing [22,23,24].

Overall, survival rate improvement was seen in most cancer sites, including colorectal [25]. The present results show that the survival rates of colorectal cancer patients in Lithuania (51.5%) are worse than the rates observed in Northern (Norway, 63.4% and Sweden, 60.3%) and Central European countries (Germany, 61.4% and Denmark, 63% colon and 65% rectal cancer) over similar time periods [5,14,26,27] and other Western countries (e.g., United States, 64.1% and United Kingdom, 53.1%) [28,29].

Although mortality is decreasing, the incidence of colorectal cancer is rising. It is known that colorectal cancer correlates with economic status and lifestyle choices, including the consumption of fatty fast food, smoking, a lack of physical activity and obesity [25]. This Westernization of lifestyle choices is becoming more common in Lithuania today. Moreover, the incidence of colorectal cancer cases has been declining in some Western countries, including the United States, France and Australia [29]. Although the difference is often attributed to the adoption of a Western lifestyle and the long-term effects of screening for colorectal cancer, no concluding explanation currently exists [29,30]. Although colorectal cancer incidence is lower in less developed regions of the world, mortality is higher, with the highest mortality rate in Eastern Europe (15.2:100.000) [4].

We have also noticed changes in treatment over time. In Lithuania, minimal invasive treatment is gaining more popularity and, since 2010, the penetration of laparoscopic surgery is more than 30–40%. Since 2010, complete mesocolic excision became a standard of care in colon cancer treatment. Furthermore, intensified treatment of advanced colorectal cancer with distant spread has also increased survival worldwide [31]. Differently from other studies, we did not see a survival improvement in advanced diseases with distant spread over the years (it has risen from 4 to 6% only). There could be a few explanations for this lack of improvement. First, supplementary nutrition is not reimbursed in Lithuania, and most patients are unfit for aggressive systemic treatment. Second, patients have refused the treatment, stating that it will not prolong their survival. Third, there is a lack of the newest systemic therapy agents on the Lithuanian market (due to difficulties with reimbursement) and the only possibility of receiving novel treatment is under international clinical trial. Contrary to another study [24], rectal cancer patients had a significantly better 5-year survival rate (66.2%) compared to colon cancer patients (52.4%; *p* ≤ 0.05). In our study, the 5-year relative survival rate was slightly higher than 50% for patients with colon cancer and almost 50% for patients with rectal cancer in the period between 2008–2012. The reason for this difference is that rectal cancer surgeries are still partly performed by general surgeons without MDT meeting discussions, omitting neoadjuvant treatment. The centralization of colorectal cancer started only in 2018, and we do not have any results yet.

Our study is the first and largest population-based study comparing cancer survival rates in Lithuania. However, this study had several limitations. Analysis was limited by variables in the database. We believe that additional patient information, such as co-morbidities, body mass index, lifestyle, multiple treatment modalities used and socioeconomic status, would have enabled analysis that was more thorough. In addition, we did not exclude the appendiceal cancers from the analysis, which represent a more distinct entity and often have different histology. However, these cancers consist in very few cases.

## 5. Conclusions

To conclude, we have shown that outcomes of colorectal cancer in Lithuania are comparable to international standards. Although survival rates variated in colorectal cancer patients according to disease stages and sub-sites, increased survival was evident for all patients, especially in patients with a localized disease. The 5-year survival was 78.6%, while survival estimates for advanced cancer remained low at 6.6%. Earlier detection and better treatment were likely the causes of these changes.

## Figures and Tables

**Figure 1 jcm-09-04038-f001:**
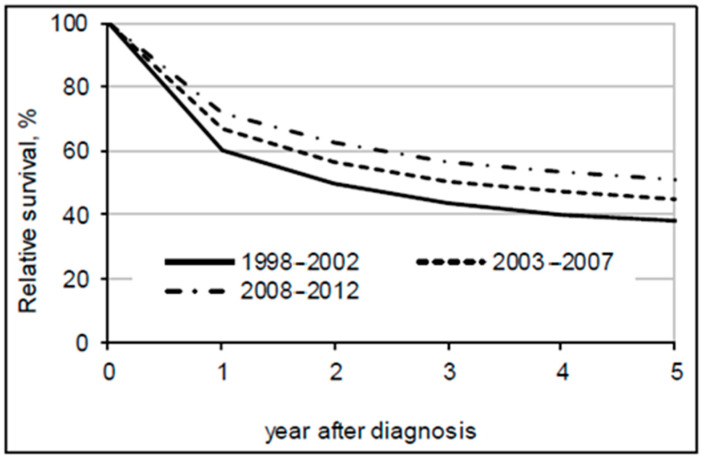
Five-year relative survival of colorectal cancer patients by period of diagnosis.

**Figure 2 jcm-09-04038-f002:**
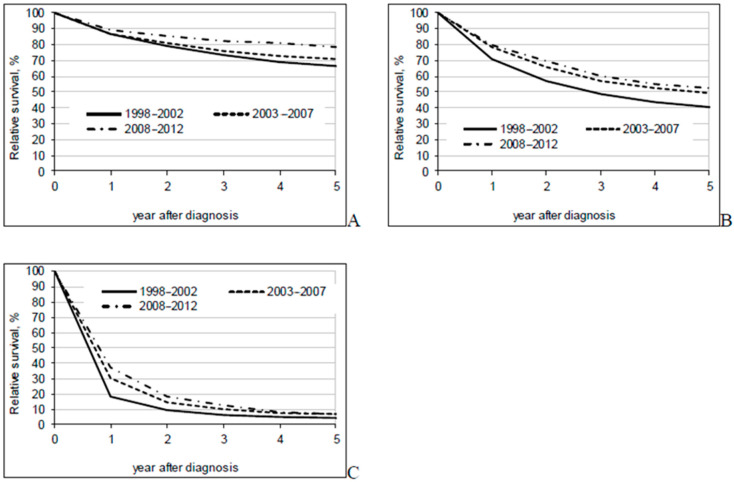
Five-year relative survival by period of diagnosis and extent of disease ((**A**)—localized; (**B**)—regional metastases; (**C**)—advanced).

**Figure 3 jcm-09-04038-f003:**
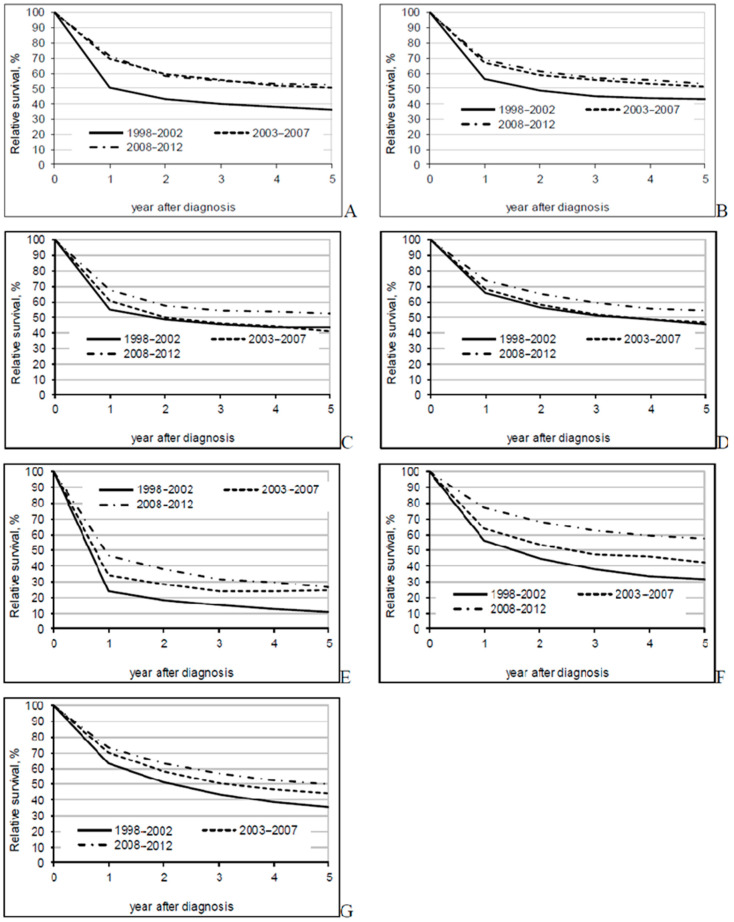
Five-year relative survival by period of diagnosis and extent of disease (**A**)—caecum and appendix; (**B**)—right colon; (**C**)—transversum; (**D**)—left colon; (**E**)—other; (**F**)—rectosigmoid; (**G**)—rectum).

**Table 1 jcm-09-04038-t001:** Distribution of the colorectal cancer cases considered in the analysis by period of diagnosis, subsite and extent of disease. Number of cases and percentage.

	1998–2002		2003–2007		2008–2012		Change *, %
	Cases	%	Cases	%	Cases	%	
Stage							
Localized tumor T1-T4	2236	33.88	2824	40.46	3106	41.97	38.91
Tumor with regional spread N+	2189	33.17	1968	28.20	1947	26.31	−11.06
Advanced cancer with distant spread	1762	26.70	1687	24.17	1489	20.12	−15.49
Missing	413	6.26	500	7.16	859	11.61	107.99
Subsite							
Caecum and appendix (C180, C181)	503	7.62	459	6.58	526	7.11	4.57
Right colon (C182, C183)	604	9.15	685	9.82	814	11.00	34.77
Transverse colon (C184)	368	5.58	351	5.03	329	4.45	−10.60
Left colon (C185, C186, C187)	1741	26.38	1889	27.07	2107	28.47	21.02
Other (C188, C189)	194	2.94	288	4.13	283	3.82	45.88
Rectosigmoid (C19)	376	5.70	450	6.45	507	6.85	34.84
Rectum (C20)	2814	42.64	2857	40.94	2835	38.31	0.75
All cases	6600	100.00	6979	100.00	7401	100.00	12.14

* percent change in the number of cases compared to the period 1998–2008.

**Table 2 jcm-09-04038-t002:** Five-year relative survival and 95% confidence interval by period of diagnosis, subsite and extent of disease.

	1998–2002	2003–2007	2008–2012	Change *, %
Stage				
Tumor localized T1-T4	66.07 (63.45–68.62)	70.84 (68.39–73.23)	78.60 (76.33–80.80)	12.53
Advanced tumor with regional spread N+	40.25 (37.98–42.54)	49.24 (46.57–51.89)	52.36 (49.57–55.14)	12.11
Advanced cancer with distant spread	4.53 (3.62–5.60)	6.61 (5.40–7.99)	6.77 (5.47–8.25)	2.24
Missing	26.74 (22.56–31.16)	22.36 (18.39–26.66)	29.02 (25.02–33.21)	2.28
Subsite				
Caecum and appendix (C18.0, C18.1)	35.71 (31.05–40.49)	50.38 (44.65–56.04)	52.13 (46.60–57.59)	16.42
Right colon (C18.2, C18.3)	43.22 (38.58–47.88)	51.33 (46.54–56.08)	52.95 (48.38–57.47)	9.73
Transverse colon (C18.4)	43.53 (37.78–49.30)	41.20 (35.12–47.38)	52.39 (45.73–58.93)	8.86
Left colon (C18.5, C18.6, C18.7)	45.77 (43.01–48.52)	46.65 (43.92–49.37)	54.12 (51.32–56.89)	8.35
Other (C18.8, C18.9)	10.84 (8.00–14.20)	24.34 (18.49–30.84)	26.40 (20.60–32.72)	15.56
Rectosigmoid (C19)	31.62 (26.47–37.01)	42.39 (36.76–48.07)	57.26 (51.26–63.10)	25.64
Rectum (C20)	35.61 (33.62–37.63)	44.49 (42.23–46.74)	49.59 (47.32–51.86)	13.98
All cases	37.86 (36.54–39.19)	45.04 (43.60–46.48)	51.13 (49.67–52.58)	13.27

* percent units change in survival compared to the period 1998–2002.

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
