# Peer review of "Improvement of Survival over Time for Colorectal Cancer Patients: A Population-Based Study"

_jcm, 2020, doi:10.3390/jcm9124038_

Round 1
Reviewer 1 Report
This study was conducted to analyse mortality and survival of CRC patients in Lithuania based on the cancer registry between 1998 and 2012. Thye observed an increase in new cases of 12.1% but the result is not expressed as a rate per population or adjusted. Localised increased from 33.9 to 42% with concomitant decreases in advanced states. 5y survival increased from 37.9% to 51.5% regardless of stage and subsite.
The title refers to subsite and stage but these did not relate to differences in survival so these would ideally be removed from the title which could better focus on the improvement seen. It is worth publishing these findings.
The Introduction should include some information on the status of screening in Lithuania and how it has changed with time. Transfer information from methods.
Methods should include information on how stage was gathered and recorded and the results of auditing such for quality. Do they believe they captured all incident CRC in that time for the country?
Results: The changes in absolute case numbers over time is informative but these data are not presented according to changes in popultion size nor are they adjusted for age and gender. Figs 1 and 2 and Table 2 require statistics to compare differences. Confidence intervals are need for Table 2.
As survival improved within stage and as earlier stage increased in proportion we can conclude that both earlier detection and better treatment were likely to cause this. The description of the state of care in Lithuania is good.
Author Response
A point-by-point response to the reviewer's comments:
Dear Editor,
Thank you for your letter and constructive comments concerning our manuscript entitled “ Improvement of Survival Over Time for Colorectal Cancer Patients by Anatomical Sub-Site and Stage: A Population-Based Study”. The paper was revised substantially. Following changes have been made. They are as follows, revised paragraphs, sentences, words are below:
Reviewer #1 This study was conducted to analyse mortality and survival of CRC patients in Lithuania based on the cancer registry between 1998 and 2012. Thye observed an increase in new cases of 12.1% but the result is not expressed as a rate per population or adjusted. Localised increased from 33.9 to 42% with concomitant decreases in advanced states. 5y survival increased from 37.9% to 51.5% regardless of stage and subsite.
The title refers to subsite and stage but these did not relate to differences in survival so these would ideally be removed from the title which could better focus on the improvement seen. It is worth publishing these findings.
Thank you for your suggestion – the title changed accordingly.
The Introduction should include some information on the status of screening in Lithuania and how it has changed with time. Transfer information from methods. Information on screening state moved from Methods to Introduction.
Methods should include information on how stage was gathered and recorded and the results of auditing such for quality. Do they believe they captured all incident CRC in that time for the country? The staging information is mentioned in Methods line 72-78. All the data is inserted by the treating physician upon his discretion. All the physicians must fill the specific form when the cancer is diagnosed. Unfortunately, we do not have any auditing body.
Results: The changes in absolute case numbers over time is informative but these data are not presented according to changes in popultion size nor are they adjusted for age and gender. Figs 1 and 2 and Table 2 require statistics to compare differences. Confidence intervals are need for Table 2. We cannot completely agree with this comment. Analysis of the trends in incidence is out of scope of our study - we aimed to analyse survival of colorectal cancer patients. In Table 1 data on cancer cases by time period, sub-site and extent of disease is given to show characteristics of the study group. According to the reviewer recommendation, we added confidence intervals in Table 2
As survival improved within stage and as earlier stage increased in proportion we can conclude that both earlier detection and better treatment were likely to cause this. The description of the state of care in Lithuania is good. Thank you for your comment. This added to the conclusions.
Thank You very much indeed.
Audrius Dulskas, MD, PhD
Reviewer 2 Report
This is a well written study by analyzing the cancer registry data in Lithuania to see the changes in the incidence, mortality and survival from 1998-2012. 1) Graphs in figure 3 for E,F,G are missing even though they were mentioned in the subscript. 2) More variables should have been used like change in comorbidities, place of surgeries(community vs tertiary hospitals, age of diagnosis (is the incidence increasing at younger age), type of surgeries, chemotherapy agents used over time to get more meaningful data from the registry. They study does show that the incidence increased and the survival for the advanced stages did not change much but no other impactful conclusions could be made.Author Response
A point-by-point response to the reviewer's comments:
Dear Editor,
Thank you for your letter and constructive comments concerning our manuscript entitled “ Improvement of Survival Over Time for Colorectal Cancer Patients by Anatomical Sub-Site and Stage: A Population-Based Study”. The paper was revised substantially. Following changes have been made. They are as follows, revised paragraphs, sentences, words are below:
Reviewer #2
This is a well written study by analyzing the cancer registry data in Lithuania to see the changes in the incidence, mortality and survival from 1998-2012. 1) Graphs in figure 3 for E,F,G are missing even though they were mentioned in the subscript. Missing data inserted. Thank you for the remark.
2) More variables should have been used like change in comorbidities, place of surgeries (community vs tertiary hospitals, age of diagnosis (is the incidence increasing at younger age), type of surgeries, chemotherapy agents used over time to get more meaningful data from the registry. They study does show that the incidence increased and the survival for the advanced stages did not change much but no other impactful conclusions could be made. Thank you for your comment. Our Cancer Registry has some information on cancer morphology, stage, treatment: surgery, radiotherapy, chemotherapy and so on. However, it still lacks the comorbidities, place of surgery (more than 80% of surgeries are performed in six major hospitals), type of specific treatment with agents and so on. All the information we could retrieve from the registry is provided in our manuscript.
Thank You very much indeed.
Audrius Dulskas, MD, PhD